# A Deep Learning Approach for Dynamic Survival Analysis with Competing Risks

## Abstract

Currently available survival analysis methods are limited in their ability to deal with complex, heterogeneous, and longitudinal data such as that available in primary care records, or in their ability to deal with multiple competing risks. This paper develops a novel deep learning architecture that flexibly incorporates the available longitudinal data comprising various repeated measurements (rather than only the last available measurements) in order to issue dynamically updated survival predictions for one or multiple competing risk(s). Unlike existing works in the survival analysis on the basis of longitudinal data, the proposed method learns the time-to-event distributions without specifying underlying stochastic assumptions of the longitudinal or the time-to-event processes. Thus, our method is able to learn associations between the longitudinal data and the various associated risks in a fully data-driven fashion. We demonstrate the power of our method by applying it to real-world longitudinal datasets and show a drastic improvement over state-of-the-art methods in discriminative performance. Furthermore, our analysis of the variable importance and dynamic survival predictions will yield a better understanding of the predicted risks which will result in more effective health care.

## 1 Introduction

Survival analysis informs our understanding of the relationships between the (distribution of) first hitting times of events (such as death, onset of a certain disease, etc.) and the covariates, and enables us to issue corresponding risk assessments for such events. Clinicians use survival analysis to make screening decisions or to prescribe treatments, while patients use the information about their clinical risks to adjust their lifestyles in order to mitigate such risks. However, designing the best clinical intervention for a patient is a daunting task, as the appropriate level of interventions or the corresponding outcome often depend on whether this patient is susceptible to or suffers from *competing risks*. For example, studies in (Koene et al. (2016)) have shown that various treatments, such as chemotherapy, for breast cancer increase the risk of a cardiovascular events. To refer to the same example in cystic fibrosis (CF), the decision on lung transplantation, which is particularly recommended for patients with end-stage respiratory failure, must jointly account for deaths from other CF-associated failures (e.g., CF-associated liver failure) since they share a number of risk factors (Kobelska-Dubiel et al. (2014)).

Meanwhile, as a growing number of electronic health records (EHRs) have been deployed in hospitals[1], modern clinical data in EHR often comprises longitudinal measurements; especially with chronic diseases, patients are followed up over the span of years, usually as part of regular physical examinations. Information contained in these longitudinal (follow-up) measurements is of significant importance such that interpreting these measurements in their historical context can offer an explanation for how the underlying process of clinical events progresses (Rizopoulos et al. (2017); Suresh et al. (2017)). For example, forced expiratory volume ($FEV_1$), and its development, is a crucial biomarker in assessing the severity of CF as it allows clinicians to describe the progression of the disease and to anticipate the occurrence of respiratory failures (Nkam et al. (2017); Li et al. (2017)). Therefore, to provide a better understanding of disease progression, it is essential to incorporate longitudinal measurements of biomarkers and risk factors into a model. Rather than

---

[1]EHRs are deployed in more than 75% of hospitals in the United States, according to the recent data brief by the Offcie of National Coordinator (ONC) (Henry et al. (2016)).

discarding valuable information recorded over time, this allows us to make better risk assessments on the clinical events of interest.

In light of the discussion above, we design a deep neural network for dynamic survival analysis with competing risks, which we call *Dynamic-DeepHit*, that learns, on the basis of the available longitudinal measurements, a flexible data-driven distribution of first hitting times of various events of interest. An important aspect of our method is that it naturally handles situations in which there are multiple competing risks where more than one type of event plays a role in the survival setting. To enable survival analysis with competing risks, Dynamic-DeepHit employs a network architecture that consists of a single shared subnetwork and a family of cause-specific subnetworks. We incorporate historical information of each patient using a recurrent neural network (RNN) structure in the shared subnetwork, which allows our method to update *personalized* and *cause-specific* risk predictions as additional measurements are collected in a fully dynamic fashion. We conduct a set of experiments on real-world datasets showing that our model outperforms state-of-the-art survival models in discriminating individual risks for different causes. In addition, we provide variable importance and dynamic risk predictions of our method to yield clinical usefulness in interpreting the associations between covariates and the survival.

**Related works:** Since the Cox proportional hazard model (Cox (1972)) was first introduced, a variety of methods have been developed for survival analysis, ranging from statistical models to deep learning techniques (Fine & Gray (1999); Ishwaran et al. (2008); Lee & Whitmore (2010); Yu et al. (2011); Lee et al. (2018); Luck et al. (2017); Katzman et al. (2016); Alaa & van der Schaar (2017); Bellot & van der Schaar (2018)). Especially, deep networks have been shown to achieve significantly improved performance in survival analysis (Katzman et al. (2016); Luck et al. (2017); Alaa & van der Schaar (2017); Bellot & van der Schaar (2018); Lee et al. (2018)) owing to the ability to represent complicated associations between features and outcomes. However, all of these methods provide static survival analysis: they use only the current information to perform the survival predictions and most of the works focus on a single risk rather than multiple risks. In this paper, we extend the idea of (Lee et al. (2018)), which finds the estimated joint distribution of the first hitting time and competing events utilizing a deep network, into the longitudinal setting where repeated measurements are available.

Few methods have been developed to use the longitudinal time-to-event data in order to enable *dynamic survival analysis* under *competing risks*. The first strand of literatures includes landmarking (Heagerty & Zheng (2005); Zheng & Heagerty (2005); van Houwlingen (2007); van Houwelingen & Putter (2008)) and joint models (Henderson et al. (2000); Ibrahim et al. (2010); Tsiatis & Davidian (2004); Brown et al. (2005); Barrett & Su (2017)). Landmarking refrains from modeling the time-dependent aspect of longitudinal variables and, instead, obtains survival probabilities from the survival model fitted to subjects who are still at risk at the time points of interest (i.e., landmarking times). On the other hand, joint models explicitly model the longitudinal process and leverage their predictions as inputs in a separate survival process used for predicting survival probabilities. However, both approaches, and their variations (van Houwelingen & Putter (2008); Tsiatis & Davidian (2004); Brown et al. (2005); Barrett & Su (2017)), make strong assumptions about the underlying stochastic models for the survival process (in landmarking) or for both the longitudinal and survival processes (in joint models). Hence, model mis-specifications (e.g., typically, a linear mixed model and a Cox proportional hazard model) limit their ability to learn and infer complex interactions between covariates and survival times, which are common in many diseases with heterogeneity.

The second strand models the longitudinal data using variants of RNNs (Choi et al. (2016a;b); Razavian et al. (2016); Lipton et al. (2016)) and avoids the need for explicit model specifications, which results in performance gain in terms of predictive accuracy. However, these models can not properly cope with time-to-event data where the goal is to find the probability of the first hitting event occurring at different times of our interest. Instead, given longitudinal measurements, they view making risk predictions of one or multiple event(s) as solving a single or multiple label(s) classification problem at each time stamp, e.g., whether an event occurs or not at the measurement time (Choi et al. (2016a;b)), within a predefined time-window (Razavian et al. (2016)), or at the end of the available longitudinal measurements (Lipton et al. (2016)). To our best knowledge, this paper is the first to investigate a deep learning approach for dynamic time-to-event analysis with competing risks on the basis of repeated measurements (longitudinal time-to-event data).

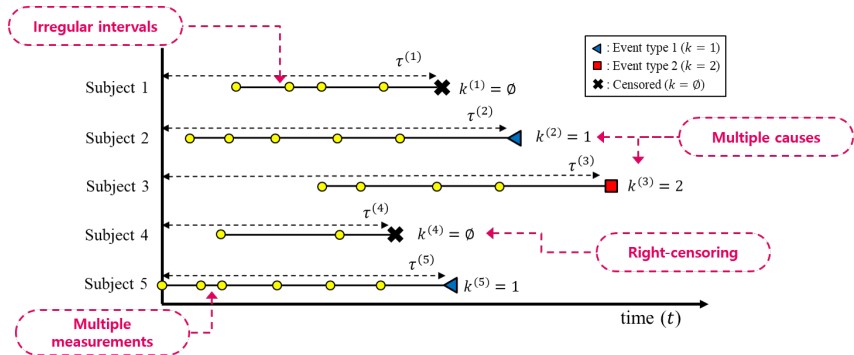

Figure 1: Illustration of longitudinal survival data. Colored dots indicate the times at which longitudinal measurements are observed, while the end point markers denote the event type or censoring.

## 2 BACKGROUND

### 2.1 LONGITUDINAL TIME-TO-EVENT DATA

We consider a dataset $\mathcal{D} = \{(\mathcal{X}^i, \tau^i, k^i)\}_{i=1}^N$ comprising time-to-event (survival) data for $N$ subjects who have been followed up for a certain amount of time. For each subject $i$, $\mathcal{X}^i$ is a set of longitudinal observations including both static and time-varying covariates, $\tau^i = \min(T^i, C^i)$ is the time information with $T^i \in \mathcal{T}$ indicating the time-to-event and $C^i \in \mathcal{T}$ indicating the time-to-censoring, and $k^i \in \mathcal{K}$ being the event or censoring that occurred at $\tau^i$.

Define $\mathcal{X}^i(t) = \{\mathbf{x}^i(t_j^i) : 0 \leq t_j^i \leq t \text{ for } j = 1, \cdots, M^i\}$ a set of longitudinal observations until time $t$ where $\mathbf{x}^i(t_j)$ is covariates recorded at time $t_j$. Here, we distinguish notations between time stamps $j = 1, \cdots, M^i$ and the corresponding actual times $t_j^i = t_1^i, \cdots, t_{M^i}^i$, since measurements are not necessarily observed at regular intervals. Then, we use $\mathcal{X}^i = \mathcal{X}^i(t_{M^i}^i)$ to denote a whole set of longitudinal observations available for subject $i$ until the last measurement time $t_{M^i}^i$ of that subject for notational simplicity. A set of possible survival times is denoted as $\mathcal{T} = \{0, 1, \cdots, T_{\max}\}$ with $T_{\max}$ being a predefined maximum time horizon, where we treat survival time as discrete[2] (e.g., a resolution of one month) and the time horizon as finite (e.g., no patients lived longer than 100 years). A set of possible events is $\mathcal{K} = \{\varnothing, 1, 2, \cdots, K\}$, with $\varnothing$ denoting *right-censoring* as survival data is frequently right-censored due to subjects being lost to follow-up. We assume that every subject experiences exactly one event among $K \geq 1$ possible events of interest within $\mathcal{T}$. This includes cause-specific deaths due to CF, where deaths from other causes are competing risks for death due to respiratory failure (Gooley et al. (1999)). Figure 1 depicts a time-to-event dataset comprising histories of longitudinal measurements with competing risks, where subjects are aligned based on the synchronization event. The aforementioned characteristics are highlighted with annotations.

### 2.2 CUMULATIVE INCIDENCE FUNCTION

The cause-specific cumulative incidence function (CIF) is key to survival analysis under the presence of competing risks. As defined in (Fine & Gray (1999)), the CIF expresses the probability that a particular event $k^* \in \mathcal{K}$ occurs on or before time $\tau^*$ conditioned on the history of longitudinal measurements $\mathcal{X}^*$. The fact that longitudinal measurements have been recorded up to $t_{M^*}^*$ implies survival of the subject up to this time point. Thus, the CIF is defined as follows:

$$F_{k^*}(\tau^*|\mathcal{X}^*) \triangleq P(T \leq \tau^*, k = k^*|\mathcal{X}^*, T > t_{M^*}^*) = \sum_{m \leq \tau^*} P(T = m, k = k^*|\mathcal{X}^*, T > t_{M^*}^*). \quad (1)$$

Whenever a new measurement is recorded for this subject at time $t > t_{M^*}^*$, we can update (1) accounting for that information in a dynamic fashion.

---

[2]Discretization is performed by transforming continuous-valued times into a set of contiguous time intervals, i.e., $T = \tau$ implies $T \in [\tau, \tau + \delta t)$ where $\delta t$ implies the resolution.

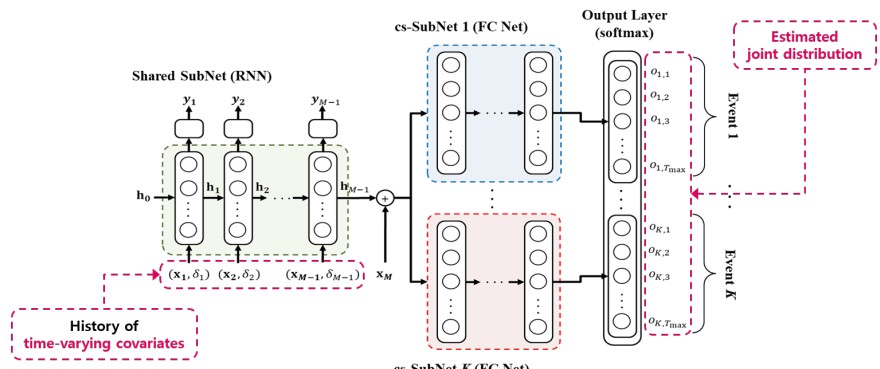

Figure 2: The Dynamic-DeepHit architecture with $K$ competing risks.

Similarly, the survival probability of a subject at time $\tau^*$ given $\mathcal{X}^*$ (i.e., the probability that a subject does not experience any event before or at time $\tau^*$) can be derived by

$$S(\tau^*|\mathcal{X}^*) \triangleq P(T > \tau^*|\mathcal{X}^*, T > t^*_{M^*}) = 1 - \sum_{k \neq \varnothing} F_k(\tau^*|\mathcal{X}^*). \tag{2}$$

Our goal is to estimate the CIF, $\hat{F}_{k^*}(\tau^*|\mathcal{X}^*)$, from the dataset $\mathcal{D}$, in order to analyze the cause-specific risk given the history of observations and to issue dynamic risk predictions.

## 3 PROPOSED METHOD

In this section, we describe our network architecture for survival analysis with competing risks on the basis of longitudinal measurements. We seek to train the network to learn an estimate of the joint distribution of the first hitting time and competing events given the longitudinal observations. This estimated distribution is then used to estimate (1) and (2).

For illustration, we redefine $\mathcal{X}^i = (\mathbf{X}^i, \Delta^i)$ where $\mathbf{X}^i = \{\mathbf{x}^i_1, \mathbf{x}^i_2 \cdots, \mathbf{x}^i_{M^i}\}$ with $\mathbf{x}^i_j = \mathbf{x}^i(t^i_j)$ and $\Delta^i = \{\delta^i_1, \delta^i_2 \cdots, \delta^i_{M^i}\}$, which is a set of time intervals between two adjacent measurements. Here, $\delta^i_j$ implies the actual amount of time that has elapsed until the next measurement, i.e. $\delta^i_j = t^i_{j+1} - t^i_j$ for $1 \leq j < M^i$, and $\delta^i_{M^i} = 0$. Then, the training set can be redefined as $\mathcal{D} = \{(\mathbf{X}^i, \Delta^i, \tau^i, k^i)\}^N_{i=1}$.

### 3.1 NETWORK ARCHITECTURE

Dynamic-DeepHit[3] is a sequence to time-to-event network, which consists of two subnetworks: a shared subnetwork that handles the history of longitudinal measurements and make step-ahead predictions of time-varying covariates, and a set of cause-specific subnetworks which estimates the joint distribution of the first hitting time and competing events. Figure 2 illustrates the overall architecture which comprises a shared subnetwork and $K$ cause-specific subnetworks. As the multi-task learning has been successful across different applications (Collobert & Weston (2008); Deng et al. (2013); Ramsundar et al. (2015)), we expect joint optimization of subnetworks to help the overall network capture associations between the time-to-event under competing risks and the history of longitudinal measurements. Throughout this subsection, we omit the dependence on $i$ for ease of notation.

**Shared Subnetwork:** The shared subnetwork employs a dynamic RNN structure to handle the longitudinal survival data with each subject having different numbers of measurements, observed at irregular time intervals. The RNN structure allows Dynamic-DeepHit to capture correlations both *within* and *across* the longitudinal measurements and to unravel the temporal patterns that are common to the $K$ competing risks. Formally, for each time stamp $j = 1, \cdots M - 1$, the subnetwork takes a tuple of $(\mathbf{x}_j, \delta_j)$ as an input and outputs $(\mathbf{y}_j, \mathbf{h}_j)$, where $\mathbf{y}_j$ and $\mathbf{h}_j$ indicates the step-ahead estimate of time-varying covariates, i.e., $\mathbf{x}_{j+1}$, and the hidden state at time stamp $j$, respectively.

**Cause-specific Subnetworks:** Each cause-specific subnetwork utilizes a feed-forward network to capture relationships between the cause-specific risk and the longitudinal measurements. The inputs

---

[3]Source code available at: http://github.com/ICLR2019-submission/Dynamic-DeepHit

include the hidden states of the shared subnetwork and the measurements at the last observation. This gives the subnetworks access to the learned common representation of the longitudinal history, which has progressed along with the trajectory of the past longitudinal measurements, as well as the latest observation. Formally, the $k$-th cause-specific subnetwork takes as input the pair $(\mathbf{x}_M, \mathbf{h}_{M-1})$ and outputs a vector, $f_{c_k}(\mathbf{x}_M, \mathbf{h}_{M-1})$.

**Output Layer:** Dynamic-DeepHit employs a soft-max layer in order to summarize the outcomes of each cause-specific subnetwork, $f_{c_1}(\cdot), \cdots, f_{c_K}(\cdot)$, and to map into one output vector. Overall, our method produces the estimated joint distribution of the first hitting time and competing events. In particular, given a subject with $\mathcal{X}^*$, each output node represents the probability of having event $k$ at time $\tau$, i.e. $o^*_{k,\tau} = \hat{P}(T = \tau, k = k|\mathcal{X}^*)$. Therefore, we can define the estimated CIF for cause $k^*$ at time $\tau^*$ as follows:

$$\hat{F}_{k^*}(\tau^*|\mathcal{X}^*) = \frac{\sum_{t^*_{M^*} < m \leq \tau^*} o^*_{k^*,m}}{1 - \sum_{k \neq \varnothing} \sum_{n \leq t^*_{M^*}} o^*_{k,n}}. \tag{3}$$

Note that (3) is built upon the condition that this subject has survived up to $t^*_{M^*}$.

## 3.2 Training Dynamic-DeepHit

To train Dynamic-DeepHit, we minimize $\mathcal{L}_{\text{total}}$ that is specifically designed to handle longitudinal measurements and right-censoring. The total loss function is the sum of three terms:

$$\mathcal{L}_{\text{total}} = \mathcal{L}_1 + \mathcal{L}_2 + \mathcal{L}_3,$$

where $\mathcal{L}_1$ is the negative log-likelihood of the joint distribution of the first hitting time and events, which is necessary to capture the first hitting time in the right-censored data, and $\mathcal{L}_2$ and $\mathcal{L}_3$ are utilized to enhance the overall network. More specifically, $\mathcal{L}_2$ combines cause-specific ranking losses to concentrate the network on discriminating estimated individual risks for each cause, and $\mathcal{L}_3$ incorporates the prediction error on trajectories of time-varying covariates to regularize the network. In Appendix C, we demonstrate the effect of including each of the losses on the performance.

**Log-likelihood Loss:** The first loss function is the negative log-likelihood of the joint distribution of the first hitting time and corresponding event considering the right-censoring (Lee & Whitmore (2006)), conditioned on the measurements recorded until the last observation. This is an extension to the survival setting with $K$ competing risks on the basis of longitudinal measurements. More specifically, for a subject who is not censored, it captures both the event that occurs and the time at which the event occurs, while for a subject who is censored, it captures the time at which the subject is censored (lost to follow-up); see (Lawless (2002)). We define the log-likelihood loss as follows:

$$\mathcal{L}_1 = - \sum_{i=1}^{N} \left[ \mathbf{1}_{k^i \neq \varnothing} \cdot \log \left( \frac{o^i_{k^i,\tau^i}}{1 - \sum_{k \neq \varnothing} \sum_{n \leq t^i_{M^i}} o^i_{k,n}} \right) + \mathbf{1}_{k^i = \varnothing} \cdot \log \left( 1 - \sum_{k \neq \varnothing} \hat{F}_k(\tau^i|\mathcal{X}^i) \right) \right]. \tag{4}$$

**Ranking Loss:** To fine-tune the network, we utilize a ranking loss function which adapts the idea of concordance (Harrell et al. (1982)): a subject who dies at time $\tau$ should have a higher risk at time $\tau$ than a subject who survived longer than $\tau$. However, the longitudinal measurements of subjects can begin at any point in their lifetime or disease progression (Ranganath et al. (2016)), and this makes direct comparison of the risks at different time points difficult to assess. Thus, we compare the risks of subjects at times elapsed since their last measurements, that is, for subject $i$, we focus on $s^i = \tau^i - t^i_{M^i}$ instead of $\tau^i$. Define a pair $(i, j)$ an *acceptable pair* for event $k$ if subject $i$ experiences event $k$ at time $s^i$ while the other subject $j$ does not experience any event, including censoring, until $s^i$ (i.e., $s^j > s^i$). Then, the estimated CIF satisfies the concordance if $\hat{F}_k(s^i + t^i_{M^i}|\mathcal{X}^i) > \hat{F}_k(s^i + t^j_{M^j}|\mathcal{X}^j)$. Formally, we define the ranking loss as follows:

$$\mathcal{L}_2 = \sum_{k=1}^{K} \alpha_k \sum_{i \neq j} A_{k,i,j} \cdot \eta \left( \hat{F}_k(s^i + t^i_{M^i}|\mathcal{X}^i), \hat{F}_k(s^i + t^j_{M^j}|\mathcal{X}^j) \right),$$

where $A_{k,i,j} \triangleq \mathbf{1}_{k^i = k, s^i < s^j}$ is an indicator for acceptable pairs $(i, j)$ for event $k$, $\alpha_k \geq 0$ is a hyper-parameter chosen to trade off ranking losses of the $k$-th competing event, and $\eta(a, b)$ is a differentiable loss function. For convenience, we set $\alpha_k = \alpha$ for $k = 1, \cdots, K$ and the loss function

$\eta(a, b) = \exp(-\frac{a-b}{\sigma})$. Incorporating $\mathcal{L}_2$ into the total loss function penalizes incorrect ordering of pairs and encourages correct ordering of pairs with respect to each event.

**Prediction Loss:** Longitudinal measurements on time-varying covariates, such as the trajectory of biomarkers and the presence of comorbidities over time, may be highly associated with the occurrence of clinical events. Thus, we introduce an auxiliary task in the shared subnetwork, which is making step-ahead predictions on covariates of our interest, to find hidden representations of historical information and to regularize the overall network. The prediction loss is defined as

$$\mathcal{L}_3 = \beta \cdot \sum_{i=1}^{N} \sum_{m=0}^{M_i-1} \zeta(\mathbf{x}_{m+1}^i, \mathbf{y}_m^i), \tag{5}$$

where $\mathbf{y}_m^i$ is the step-ahead prediction for $\mathbf{x}_{m+1}^i$, $\beta \geq 0$ is a hyper-parameter, and $\zeta(\mathbf{a}, \mathbf{b}) = \sum_{d \in \mathcal{I}} \zeta_d(a_d, b_d)$ where $\mathbf{a}$ and $\mathbf{b}$ are $d_x$-dimensional vectors, and $\zeta_d(a_d, b_d) = |a_d - b_d|^2$ for continuous and $\zeta_d(a_d, b_d) = a_d \log b_d + (1 - a_d) \log(1 - b_d)$ for binary covariates. Here, we select $\mathcal{I}$ as a set of time-varying covariates (e.g., biomarkers) on which we focus the network to be regularized.

### 3.3 DISCUSSION ON THE SCALABILITY

For accurate estimation of CIFs, it is desirable to have the time interval for the time horizon discretization (i.e., defining $\mathcal{T}$ in Section 2) be fine rather than coarse since it maintains more information on time-to-event/censoring. However, this might cause scalability issue in Dynamic-DeepHit. In particular, the proposed network requires a larger number of nodes in the output layer to handle $\mathcal{T}$ with a finer time interval, and the network can become over-fitted due to a small number of samples in the dataset, which is often the case in medicine. To prevent the proposed network from over-fitting, we utilize 1) early stopping based on the performance metric of our interest (i.e., discriminative performance) and 2) $L_1$ regularization over weights in the cause-specific subnetwork and the output layer. We show in the experiments that Dynamic-DeepHit achieves a significant performance gain with a fine time interval; see details in the following section.

## 4 EXPERIMENTS

Throughout the experiments, we use two real-world medical datasets comprising two competing risks to evaluate our proposed method against the state-of-the-art benchmarks. (In addition, we reported the evaluation of our method for a real-world dataset with a single risks in Appendix E.) Results are obtained using 5 random 64/16/20 train/validation/test splits. The hyper-parameters are chosen utilizing Random Search (Bergstra & Bengio (2012)); see details in Appendix A. For the prediction loss in (5), we included all the time-varying covariates available in each dataset into $\mathcal{I}$.

### 4.1 DATASET

**Cystic Fibrosis Data:** This is a retrospective longitudinal data from the UK Cystic Fibrosis Registry, sponsored and hosted by the UK Cystic Fibrosis Trust[4]. We focused our experiments on 5,883 adult cystic fibrosis (CF) patients who were aged 18 years or older. Among the total of 5,883 patients, 605 patients (10.28%) were followed until death and the remaining 5,278 patients (89.72%) were right-censored (i.e., lost to follow-up). We divided the mortality cause into: i) 491 (8.35%) deaths due to respiratory failures and ii) 114 (1.94%) deaths due to other causes, including CF-associated liver failure. (Details on the other causes are illustrated in Appendix D.) It is important that patients who are at risk of respiratory failure be provided with a joint prognosis of mortality due to other causes in order to properly manage therapeutic interventions; lung transplantation is particularly recommended for patients with end-stage respiratory failure (Liou et al. (2001); Hofer et al. (2009)) since not only lung donors are very scarce but also it is accompanied with serious risks of post-transplant complications (Mayer-Hamblett et al. (2002)). Since CF is a genetic disease where the longitudinal measurements of patients can begin at any point in their disease progression, throughout the experiments, all patients are aligned based on their date of birth to synchronize the time to compare risk predictions made at different ages. We prescribed the set of possible survival times up to 100 years with a monthly time interval, i.e., $\mathcal{T} = \{0, 1, \cdots, 1200\}$.

---

[4]https://www.cysticfibrosis.org.uk/the-work-we-do/uk-cf-registry

**Mayo Clinic Primary Biliary Cirrhosis Data:** This data is from the Mayo Clinic trial in primary biliary cirrhosis (PBC) of the liver, which comprises of 312. Among 312 PBC patients, 169 patients (54.17%) were followed until the competing events of our interest and the remaining 143 patients (45.83%) were right-censored. Two competing events are considered in this experiment: i) 140 (44.87%) deaths and ii) 29 (9.29%) patients who underwent liver transplantation. Since not only death from liver failure is highly associated with and have many common risk factors with liver transplantation but also it hinders the death being observed during study, it is of significant importance to take liver transplantation into account as the competing risk. Throughout the experiments, all patients are aligned based on the start of the clinical study on PBC. We prescribed the set of possible survival times up to 15 years with a monthly time interval, i.e., $\mathcal{T} = \{0, 1, \cdots, 180\}$.

Missing imputations and full descriptions of the datasets are available in Appendix D.

## 4.2 BENCHMARKS

We compared Dynamic-DeepHit with the following state-of-the-art methods that account for dynamic survival analysis based on longitudinal measurements (please refer to Appendix B for details):

- **cs-Cox**: a landmarking approach utilizing the cause-specific version of the Cox Proportional Hazards (PH) Model[5],
- **RSF**: a landmarking approach utilizing the random survival forest[6] (Ishwaran et al. (2008)) with 1000 trees,
- **DeepHit**: a landmarking approach utilizing the deep neural network for competing risks proposed in (Lee et al. (2018)),
- **JM**: a joint model implemented using a Bayesian framework that uses MCMC algorithms[7] (Rizopoulos (2016)) by modeling the time-to-event data using a cause-specific Cox PH model and the longitudinal process using a linear mixed model,
- **Dynamic-Exp**: a deep neural network utilizing the same architecture with that of Dynamic-DeepHit whose output layer is modified to model the time-to-event data with the Exponential distribution.

Here, to account for the competing risks setting, the cause-specific Cox was created by fixing an event and treating the other event simply as a form of censoring; see (Haller et al. (2013)). DeepHit and Dynamic-Exp are introduced to highlight the gain of our proposed architecture.

## 4.3 DISCRIMINATIVE PERFORMANCE

In survival analysis, predictions of survival models are most commonly assessed and compared with respect to how well the survival models discriminate individual risks. To assess the discriminative performance of the various methods, we use a cause-specific time-dependent concordance index, $C_k(t, \Delta t)$, which is an extension of (Gerds et al. (2013)) adapted to the competing risks setting on the basis of longitudinal measurements.[8] In particular, $C_k(t, \Delta t)$ takes both prediction and evaluation times into account to reflect possible changes in predicted risks over time. Given the estimated CIF in (3), $C_k(t, \Delta t)$ for event $k$ is defined as

$$C_k(t, \Delta t) = P\Big( \hat{F}_k(t + \Delta t | \mathcal{X}^i(t)) > \hat{F}_k(t + \Delta t | \mathcal{X}^j(t)) \Big| T^i < T^j, k^i = k, T^i < t + \Delta t \Big), \quad (6)$$

where $t$ is the prediction time, which is the time at which survival models issue risk predictions incorporating measurements collected until that time, and $\Delta t$ is the evaluation time, which is the time elapsed since the prediction is made.

We reported the discriminative performance of survival models on the CF dataset and the PBC dataset in Table 1 and 2, respectively. On the CF dataset, our model achieves significant performance

---

[5]https://cran.r-project.org/web/packages/survival/

[6]https://cran.r-project.org/web/packages/randomForestSRC/

[7]https://cran.r-project.org/web/packages/JMbayes/

[8]The metric in (Gerds et al. (2013)) is suitable for evaluating discriminative performance at different time horizons once risk predictions are issued in the static survival setting. However, since the time horizon at which risk predictions are made is not considered, this metric cannot be directly used in the longitudinal setting; similar extensions are proposed using area under ROC curve in (Rizopoulos et al. (2017); Suresh et al. (2017))

Table 1: Comparison of $C_k(t, \Delta t)$ (mean $\pm$ std) for the CF dataset. Higher the better.

| Algorithms | | Resp. Failure | | | Other Causes | | |
|---|---|---|---|---|---|---|---|
| | | $\Delta t = 1$ | $\Delta t = 3$ | $\Delta t = 5$ | $\Delta t = 1$ | $\Delta t = 3$ | $\Delta t = 5$ |
| $t = 30$ | cs-Cox | 0.748±0.10 | 0.748±0.09 | 0.748±0.09 | 0.604±0.13 | 0.601±0.13 | 0.602±0.13 |
| | RSF | 0.935±0.01 | 0.926±0.01 | 0.925±0.01 | 0.799±0.04 | 0.791±0.05 | 0.772±0.05 |
| | DeepHit | 0.910±0.02 | 0.907±0.02 | 0.907±0.02 | 0.819±0.07 | 0.831±0.07 | 0.834±0.07 |
| | JM | 0.833±0.02 | 0.878±0.01 | 0.870±0.01 | 0.728±0.04 | 0.766±0.05 | 0.759±0.05 |
| | Dynamic-Exp. | 0.895±0.03 | 0.890±0.03 | 0.890±0.03 | 0.824±0.05 | 0.825±0.05 | 0.824±0.05 |
| | Dynamic-DeepHit | **0.946±0.01** | **0.940±0.01** | **0.939±0.01** | **0.926±0.02** | **0.919±0.03** | **0.913±0.03** |
| $t = 40$ | cs-Cox | 0.745±0.04 | 0.745±0.04 | 0.745±0.04 | 0.604±0.14 | 0.605±0.14 | 0.605±0.14 |
| | RSF | 0.886±0.02 | 0.887±0.03 | 0.885±0.03 | 0.801±0.10 | 0.772±0.05 | 0.744±0.05 |
| | DeepHit | 0.913±0.02 | 0.923±0.02 | 0.923±0.01 | 0.837±0.07 | 0.845±0.07 | 0.846±0.07 |
| | JM | 0.858±0.02 | 0.872±0.01 | 0.884±0.01 | 0.775±0.04 | 0.782±0.04 | 0.787±0.04 |
| | Dynamic-Exp. | 0.883±0.03 | 0.883±0.03 | 0.882±0.03 | 0.816±0.04 | 0.817±0.04 | 0.816±0.04 |
| | Dynamic-DeepHit | **0.944±0.03** | **0.954±0.01** | **0.954±0.01** | **0.920±0.02** | **0.918±0.03** | **0.921±0.02** |
| $t = 50$ | cs-Cox | 0.801±0.11 | 0.801±0.11 | 0.801±0.11 | 0.649±0.15 | 0.649±0.15 | 0.649±0.15 |
| | RSF | 0.895±0.01 | 0.891±0.02 | 0.889±0.02 | 0.731±0.06 | 0.763±0.03 | 0.763±0.03 |
| | DeepHit | 0.929±0.01 | 0.929±0.01 | 0.929±0.01 | 0.851±0.07 | 0.858±0.06 | 0.859±0.06 |
| | JM | 0.878±0.02 | 0.884±0.01 | 0.889±0.01 | 0.784±0.04 | 0.788±0.04 | 0.791±0.04 |
| | Dynamic-Exp. | 0.875±0.02 | 0.874±0.02 | 0.874±0.02 | 0.806±0.04 | 0.806±0.04 | 0.806±0.04 |
| | Dynamic-DeepHit | **0.958±0.01** | **0.959±0.01** | **0.959±0.01** | **0.934±0.02** | **0.939±0.02** | **0.938±0.02** |

Table 2: Comparison of $C_k(t, \Delta t)$ (mean $\pm$ std) for the PBC dataset. Higher the better.

| Algorithms | | Death | | |
|---|---|---|---|---|
| | | $\Delta t = 1$ | $\Delta t = 3$ | $\Delta t = 5$ |
| $t = 2$ | cs-Cox | 0.900±0.02 | 0.903±0.01 | 0.877±0.01 |
| | RSF | 0.912±0.03 | 0.902±0.02 | 0.874±0.02 |
| | DeepHit | 0.878±0.02 | 0.853±0.02 | 0.839±0.03 |
| | JM | 0.905±0.04 | 0.898±0.03 | 0.866±0.02 |
| | Dynamic-Exp | **0.914±0.03** | 0.907±0.03 | 0.890±0.02 |
| | Dynamic-DeepHit | 0.904±0.03 | **0.907±0.02** | **0.890±0.01** |
| $t = 4$ | cs-Cox | 0.890±0.02 | 0.875±0.02 | 0.864±0.02 |
| | RSF | 0.878±0.02 | 0.857±0.02 | 0.843±0.01 |
| | DeepHit | 0.867±0.02 | 0.841±0.01 | 0.829±0.02 |
| | JM | 0.868±0.03 | 0.838±0.02 | 0.812±0.02 |
| | Dynamic-Exp | 0.889±0.02 | 0.870±0.02 | 0.852±0.02 |
| | Dynamic-DeepHit | **0.896±0.03** | **0.896±0.02** | **0.882±0.01** |
| $t = 6$ | cs-Cox | 0.824±0.02 | 0.809±0.01 | 0.806±0.02 |
| | RSF | 0.823±0.01 | 0.828±0.02 | 0.827±0.01 |
| | DeepHit | 0.836±0.01 | 0.827±0.02 | 0.824±0.02 |
| | JM | 0.784±0.02 | 0.761±0.02 | 0.741±0.01 |
| | Dynamic-Exp | 0.857±0.02 | 0.836±0.02 | 0.824±0.03 |
| | Dynamic-DeepHit | **0.894±0.01** | **0.883±0.01** | **0.861±0.02** |

gain for all the tested predication times (in age) and evaluation times (in year) in comparison with state-of-the-art methods for both causes, especially, providing higher gain in discriminating risks of other death causes. On the PBC dataset, we only assessed the discriminative performance on the predicted risks for death; the probability of having liver transplantation is not in our interest. Even though the dataset is relatively small, Dynamic-DeepHit provided comparable performance to the best performing benchmark for evaluation times $t = 2$ (in year), while it achieved significant gain over for $t = 4$ and 6, where the proposed method can collect more measurements when making the risk predictions.

Dynamic-DeepHit benefits from the proposed architecture. First, the RNN structure in the shared subnetwork renders our method to incorporate the measurement history when making the risk predictions. This leads to performance improvement over the conventional DeepHit which discards the historical information and relies only on the last available measurement. Second, Dynamic-Exp suffers from model mis-specification (i.e., the Exponential distribution) by limiting the network to learn the complex interactions between the longitudinal measurements and the underlying survival process. Contrarily, Dynamic-DeepHit better estimates the CIFs utilizing the proposed output layer by flexibly learning the joint distribution of the first hitting time and the competing events.

### 4.4 Variable Importance via Partial Dependence

In this subsection, we utilize a post-processing statistic that can be used by clinicians to interpret predictions issued by Dynamic-DeepHit and to understand the associations of covariates and the survival. It is worth drawing a distinction between interpreting a model, versus interpreting its deci-

sion (Ribeiro et al. (2016); Avati et al. (2017)). While interpreting complex models (e.g deep neural networks) may sometimes be infeasible, it is often the case that clinicians only want explanations for the prediction made by the model for a given subject. To help interpret predictions issued by Dynamic-DeepHit, we leverage the partial dependence introduced in (Friedman (2001)) by extending it to the survival setting with competing risks.

Let $\mathcal{X}_\ell$ be a chosen target subset of the input covariates, $\mathcal{X}$, and $\mathcal{X}_{\setminus \ell}$ be its complement, i.e., $\mathcal{X}_\ell \cup \mathcal{X}_{\setminus \ell} = \mathcal{X}$. Then, we can rewrite (3) as $\hat{F}_k(\tau|\mathcal{X}) = \hat{F}_k(\tau|\mathcal{X}_\ell, \mathcal{X}_{\setminus \ell})$ to explicitly denote the dependency on the target subset. For each event $k$, the partial dependence function at time $\Delta t$, which is the time elapsed since the last measurements, can be defined as follows:

$$\gamma_k(\Delta t, \mathcal{X}_\ell) = \mathbb{E}_{\mathcal{X}_{\setminus \ell}} \left[ \hat{F}_k(t_M + \Delta t|\mathcal{X}_\ell, \mathcal{X}_{\setminus \ell}) \right] \approx \frac{1}{N} \sum_{i=1}^{N} \hat{F}_k(t_{M^i}^i + \Delta t|\mathcal{X}_\ell, \mathcal{X}_{\setminus \ell}^i), \qquad (7)$$

where $t_M$ indicates the time of the last measurement. It is worth to highlight that from (7) we can approximately assess how the estimated CIFs are affected by different values of $\mathcal{X}_\ell$ on average. As a measure of variable importance in making cause-specific risk predictions, we use the "ratio of change" in (7) by increasing the input value from the minimum ($x_{\ell,\min}$) to the maximum ($x_{\ell,\max}$) of $\mathcal{X}_\ell$, i.e., the ratio of change between $\gamma_k(\Delta t, \mathcal{X}_\ell = x_{\ell,\min})$ and $\gamma_k(\Delta t, \mathcal{X}_\ell = x_{\ell,\max})$.

In Table 3, we reported ten most influential covariates, which provide the top ten largest ratio of change, for each cause with $\Delta t = 5$ for the CF datasets. Here, positive and negative signs indicate whether the higher value of each target covariate increases (+) or decreases (-) the risk predictions. It is worth to highlight that the rankings in Table 3 are different between two causes – for example, indicator for cancer significantly influences the risk prediction for other causes while it has marginal influence on the risk prediction for respiratory failure. In addition, our method found the lung function score (FEV$_1$% predicted), nutritional status (BMI and weight), and days of intravenous (IV) antibiotics in hospital as important covariates for predicting the risks, confirming clinical findings of CF: 1) FEV$_1$% predicted is a strong surrogate for the survival, where its decrease severely increases the mortality of CF patients (Aarona et al. (2015)), 2) hospitalization periods are often considered as key risk factors for CF patients (Nkam et al. (2017)), and 3) the occurrence of malnutrition, which is often indicated by low BMI, is associated with reductions in their survival (Stephenson et al. (2013)).

Table 3: The ranking of the ten most influential covariates with $\Delta t = 5$ year. The values indicate the ratio of increase/decrease of the partial dependence function.

| Ranking | Resp. Failure | Other Causes |
|---------|---------------|--------------|
| 1 | IV Antibiotic Days (Hospital) (+1.65) | Colonic Stricture (+0.89) |
| 2 | FEV$_1$% Predicted (-0.85) | IV Antibiotic Days (Hospital) (+0.79) |
| 3 | GI Bleeding (Non Variceal) (-0.69) | Cancer (+0.44) |
| 4 | Gram-Negative (-0.68) | FEV$_1$% Predicted (-0.43) |
| 5 | HD iBuprofen (-0.66) | Gram-Negative (-0.40) |
| 6 | O$_2$ Continuous (+0.65) | GI Bleeding (Variceal) (+0.39) |
| 7 | BMI (-0.54) | O$_2$ Continuous (+0.38) |
| 8 | Weight (-0.49) | HD iBuprofen (-0.32) |
| 9 | GI Bleeding (Variceal) (+0.46) | BMI (-0.28) |
| 10 | Oral Hypoglycemic Agents (-0.44) | Pancreatitis (-0.27) |

## 4.5 DYNAMIC RISK PREDICTION

At run-time, Dynamic-DeepHit issues cause-specific risk predictions for each subject incorporating his/her medical history; we illustrate dynamic risk predictions and the trajectory of the hidden states for representative patients of the CF dataset in Figure 3 and 6. Whenever a new observation is made, Dynamic-DeepHit updates the predictions that start from 0 due to the fact that this patient is alive at the time of the measurement as defined in (3). However, the predicted risks can vary significantly depending on the new measurements. For instance, the predicted risks for the patient in Figure 3, who died from respiratory failure, were relatively higher and steeper compared to those for the patient in Figure 6, who died from other causes. This is presumably because increasing antibiotic treatment days in hospital (IV ABX hosp.) and decreasing FEV$_1$% predicted have more influence on the risk prediction for the respiratory failure, while other factors, such as colonic stricture and cancer, are important for risk predictions for the other causes. Figure 3 and 4 (b) show 2-dimensional PCA projection of the hidden states of the shared network for the corresponding patients to illustrate how the

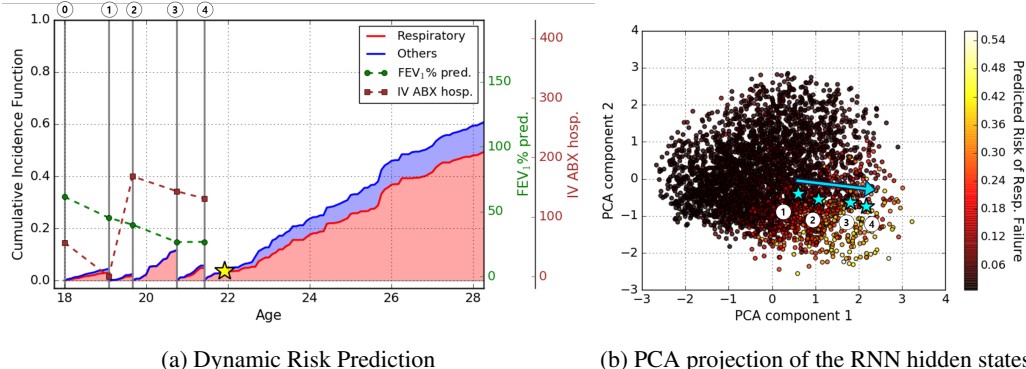

(a) Dynamic Risk Prediction        (b) PCA projection of the RNN hidden states

Figure 3: Illustration of (a) dynamic risk predictions and (b) PCA projection of the RNN hidden states for a CF patient died from respiratory failure. (a) The gray solid lines indicates the time at which new measurements are collected and the yellow star denotes the time at which the respiratory failure occurred. (b) The blue stars denote the corresponding PCA projection of the hidden states.

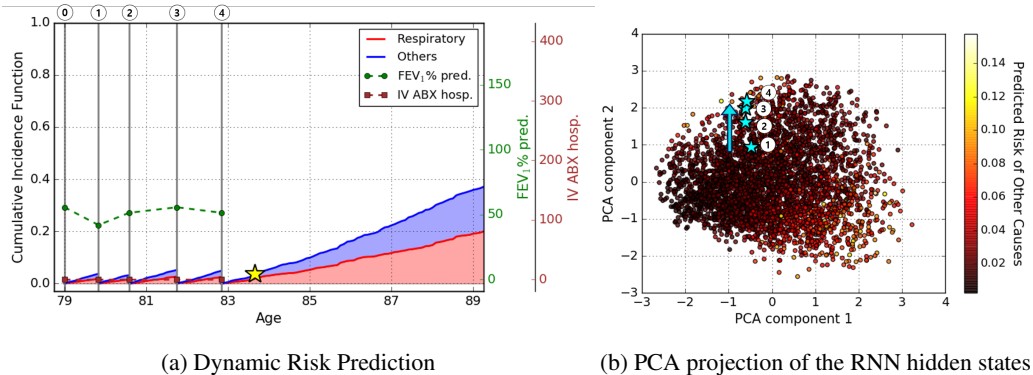

(a) Dynamic Risk Prediction        (b) PCA projection of the RNN hidden states

Figure 4: Illustration of (a) dynamic risk predictions and (b) PCA projection of the RNN hidden states for a CF patient died from other causes. (a) The gray solid lines indicates the time at which new measurements are collected and the yellow star denotes the time at which the other causes occurred. (b) The blue stars denote the corresponding PCA projection of the hidden states.

hidden state changes as new measurements are collected over time. The same PCA decomposition is used for both patients while different color maps are used to differentiate the predicted risk for the respiratory failure and that for the other causes. (Here, since the shared network takes only the previous measurements as the input, we omitted the hidden state at the first measurement, which is denoted as 0, in the figures.) We confirmed that the trajectory of the hidden states in the shared network for the patient in Figure 3 moved toward "high risk" of respiratory failure region while that for the patient in Figure 4 moved toward "high risk" of other causes region. This highlights the usefulness of the shared network as the hidden states evolve along with the history for measurements. (The dynamic risk prediction for a censored patient is provided in Appendix G.)

## 5 CONCLUSION

In this paper, we developed a novel approach, Dynamic-DeepHit, to perform dynamic survival analysis with competing risks on the basis of longitudinal data. Dynamic-DeepHit is a deep neural network which learns the estimated joint distributions of survival times and competing events, without making assumptions regarding the underlying stochastic processes. We train the network by leveraging a combination of loss functions that capture the right-censoring and the associations of longitudinal measurements, both of which are inherent in time-to-event data. We demonstrated the utility of our proposed method through experiments conducted on real-world survival datasets with competing risks, which comprise patients with follow-up measurements. The experiments show that the proposed method significantly outperforms the state-of-the-art benchmarks in terms of discriminative performance.

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

## A Hyper-parameter Optimization

The hyper-parameters, such as the coefficients, the activation functions, and the number of hidden layers and nodes of each subnetwork, are chosen utilizing Random Search (Bergstra & Bengio (2012)). The permitted values of the hyper-parameters are listed in Table 4.

Table 4: Hyper-parameters of Dynamic-DeepHit

| Block | Sets of hyper-parameters |
|---|---|
| Initialization | Xavier initialization for weight matrix |
| | Zero initialization for bias vector |
| Optimization | Adam Optimizer |
| RNN architecture | Bi-directional LSTM |
| Nonlinearity | $\{\text{ReLU, ELU, tanh}\}$ |
| Dropout | 0.6 |
| Learning rate | $\{10^{-4}, 10^{-5}\}$ |
| Mini-batch size | $\{32, 64, 128\}$ |
| No. of layers | $\{1, 2, 3\}$ |
| No. of hidden nodes | $\{50, 100, 200\}$ |
| $\alpha, \beta, \sigma$ | $\{0.1, 1, 3, 5, 10\}$ |

## B Details of the Benchmarks

We compared Dynamic-DeepHit with state-of-the-art methods that account for dynamic survival analysis under the presence of longitudinal measurements, including the joint model (Henderson et al. (2000)), survival methods under landmarking approaches (van Houwlingen (2007)), and a deep neural network that applies the similar architecture with that of our proposed network.

### B.1 Joint Model

We implement the joint model (**JM**) under a Bayesian framework that uses MCMC algorithms via R package 'JMbayes'[9]. The modeling framework for the longitudinal process is a linear mixed effects model linked to the time-to-event process, which is modeled with Cox PH model. Since the R package only supports for the survival models in the single risk setting, we constructed the cause-specific Cox for each event by fixing an event (e.g., death from respiratory cause) and treating the other event (e.g., death from other causes) simply as a form of censoring to account for the competing risks setting; this is a common approach as well-described in (Haller et al. (2013)). For the longitudinal process, we selected $FEV_1\%$ predicted for the CF dataset and serum bilirubin for the PBC dataset to model the longitudinal processes since these covariates are known as the most prognostic biomarker for corresponding diseases (Nkam et al. (2017); Shapiro et al. (1979)) and standard joint models suffer from computational limitations for modeling all time-varying covariates (Hickey et al. (2016)).

### B.2 Landmarking

We follow the description in (van Houwlingen (2007)) to construct landmarking benchmarks. More specifically, the landmarking times are chosen as the prediction times, and only patients who are at risk at these landmarking times (patients who have not experienced any event or been censored) are considered when we fit survival models at each landmarking time. Overall, the landmarking approaches are implemented utilizing the following survival models: the cause-specific version of the Cox PH Model (**cs-Cox**) vis R package 'survival'[10], random survival forests under competing risks (**RSF**) via R package 'randomForestSRC'[11] (Ishwaran et al. (2008)) with 1000 trees

---

[9]https://cran.r-project.org/web/packages/JMbayes/
[10]https://cran.r-project.org/web/packages/survival/
[11]https://cran.r-project.org/web/packages/randomForestSRC/

as a non-parametric alternative of the Cox model, and the deep neural network for competing risks (**DeepHit**) proposed by (Lee et al. (2018)) to highlight the gain of incorporating history of measurements (the RNN in the shared subnetwork is replaced with a feed-forward network and the network is trained without $\mathcal{L}_3$).

### B.3 DYNAMIC-EXP: A PARAMETRIC VERSION OF DYNAMIC-DEEPHIT

We construct a parametric version of our proposed network (**Dynamic-Exp**) by modifying the output layer to model the underlying survival process with the Exponential distribution. In particular, the output layer is replaced from a soft-max layer with multiple output nodes to a soft-plus layer with a single output node which parameterizes the Exponential distribution of the survival process for each cause. Formally, define $T_k$ the time-to-event of cause $k$; then $T_k \sim \text{Exp}(\lambda_k)$ where $\lambda_k = \text{Softplus}\big(f_{c_k}(\mathbf{x}_M, \mathbf{h}_{M-1})\big)$. (Here, the same notations in Section 2 are adopted.) Given the outputs of the network for subject $i$ as $\lambda_1^i, \cdots, \lambda_K^i$, the log-likelihood loss, $\mathcal{L}_1$, in (4) can be redefined as follows:

$$\mathcal{L}_1 = -\sum_{k=1}^{K}\sum_{i=1}^{N}\left[\mathbf{1}_{k^i=k}\cdot\log P\big(T_k=\tau^i|\mathcal{X}^i, T_k>t_{M^i}^i\big) + \mathbf{1}_{k^i\neq k}\cdot\log P\big(T_k>\tau^i|\mathcal{X}^i, T_k>t_{M^i}^i\big)\right],$$

$$= -\sum_{k=1}^{K}\sum_{i=1}^{N}\left[\mathbf{1}_{k^i=k}\cdot\log\big(\lambda_k^i\exp(-\lambda_k^i s^i)\big) + \mathbf{1}_{k^i\neq k}\cdot\big(-\lambda_k^i s^i\big)\right],$$

where $s^i = \tau^i - t_{M^i}^i$ implies the time-to-event/censoring from the last measurement time and the second equation is derived from the definition of the Exponential distribution.

This network is introduced to highlight how Dynamic-DeepHit can benefit from directly learning the joint distribution of the first hitting time and the competing events without any assumptions about the underlying survival process. Throughout the evaluation, we train Dynamic-Exp adopting the modified output layers and $\mathcal{L}_1$ via the same random search for hyper-parameter optimization that is used to train our proposed network.

## C UNDERSTANDING THE SOURCE OF GAIN

Dynamic-DeepHit is trained based on three loss functions, each of which has a different role in optimizing the overall network for the considered problem. To further understand where the gains come from, we compared the discriminative performance that is achieved when the network is trained utilizing only parts of the loss functions and when the network does not incorporate the longitudinal history. More specifically, Table 5 shows $C_k(t, \Delta t)$ for the CF dataset comparing the following variations: Dynamic-DeepHits that are trained only with $\mathcal{L}_1$, $\mathcal{L}_2$, $\mathcal{L}_1 + \mathcal{L}_2$, and $\mathcal{L}_1 + \mathcal{L}_3$, respectively. For all the comparisons, the same hyper-parameter optimization is applied.

Table 5: Comparison of $C_k(t, \Delta t)$ (mean $\pm$ std) for CF dataset with various settings.

| Algorithms | | Resp. Failure | | | Other Causes | | |
|---|---|---|---|---|---|---|---|
| | | $\Delta t = 1$ | $\Delta t = 3$ | $\Delta t = 5$ | $\Delta t = 1$ | $\Delta t = 3$ | $\Delta t = 5$ |
| $t = 30$ | $\mathcal{L}_1$ | 0.893$\pm$0.02 | 0.893$\pm$0.02 | 0.898$\pm$0.02 | 0.859$\pm$0.03 | 0.867$\pm$0.03 | 0.853$\pm$0.04 |
| | $\mathcal{L}_2$ | 0.907$\pm$0.04 | 0.880$\pm$0.08 | 0.863$\pm$0.12 | 0.868$\pm$0.10 | 0.911$\pm$0.03 | **0.915$\pm$0.03** |
| | $\mathcal{L}_1 + \mathcal{L}_2$ | 0.941$\pm$0.01 | 0.937$\pm$0.01 | 0.937$\pm$0.01 | 0.919$\pm$0.03 | 0.914$\pm$0.02 | 0.914$\pm$0.03 |
| | $\mathcal{L}_1 + \mathcal{L}_3$ | 0.899$\pm$0.01 | 0.899$\pm$0.01 | 0.903$\pm$0.01 | 0.866$\pm$0.04 | 0.865$\pm$0.05 | 0.866$\pm$0.04 |
| | $\mathcal{L}_{\text{total}}$ | **0.946$\pm$0.01** | **0.940$\pm$0.01** | **0.939$\pm$0.01** | **0.926$\pm$0.02** | **0.919$\pm$0.03** | 0.913$\pm$0.03 |
| $t = 40$ | $\mathcal{L}_1$ | 0.856$\pm$0.09 | 0.891$\pm$0.02 | 0.887$\pm$0.02 | 0.874$\pm$0.04 | 0.871$\pm$0.03 | 0.862$\pm$0.04 |
| | $\mathcal{L}_2$ | 0.846$\pm$0.19 | 0.832$\pm$0.22 | 0.852$\pm$0.18 | 0.813$\pm$0.12 | 0.803$\pm$0.12 | 0.697$\pm$0.14 |
| | $\mathcal{L}_1 + \mathcal{L}_2$ | **0.947$\pm$0.01** | 0.949$\pm$0.01 | 0.949$\pm$0.01 | 0.918$\pm$0.01 | 0.909$\pm$0.03 | 0.921$\pm$0.01 |
| | $\mathcal{L}_1 + \mathcal{L}_3$ | 0.853$\pm$0.08 | 0.887$\pm$0.01 | 0.885$\pm$0.01 | 0.816$\pm$0.06 | 0.810$\pm$0.07 | 0.813$\pm$0.07 |
| | $\mathcal{L}_{\text{total}}$ | 0.944$\pm$0.03 | **0.954$\pm$0.01** | **0.954$\pm$0.01** | **0.920$\pm$0.02** | **0.918$\pm$0.03** | **0.921$\pm$0.02** |
| $t = 50$ | $\mathcal{L}_1$ | 0.882$\pm$0.03 | 0.873$\pm$0.02 | 0.877$\pm$0.02 | 0.808$\pm$0.03 | 0.823$\pm$0.01 | 0.757$\pm$0.11 |
| | $\mathcal{L}_2$ | 0.663$\pm$0.17 | 0.593$\pm$0.29 | 0.621$\pm$0.25 | 0.884$\pm$0.04 | 0.791$\pm$0.21 | 0.823$\pm$0.13 |
| | $\mathcal{L}_1 + \mathcal{L}_2$ | 0.948$\pm$0.02 | 0.948$\pm$0.01 | 0.949$\pm$0.01 | 0.913$\pm$0.03 | 0.916$\pm$0.03 | 0.916$\pm$0.03 |
| | $\mathcal{L}_1 + \mathcal{L}_3$ | 0.860$\pm$0.04 | 0.866$\pm$0.03 | 0.870$\pm$0.02 | 0.795$\pm$0.06 | 0.803$\pm$0.06 | 0.783$\pm$0.09 |
| | $\mathcal{L}_{\text{total}}$ | **0.958$\pm$0.01** | **0.959$\pm$0.01** | **0.959$\pm$0.01** | **0.934$\pm$0.02** | **0.939$\pm$0.02** | **0.938$\pm$0.02** |

Dynamic-DeepHit improved the discriminative performance by combining two additional loss functions, $\mathcal{L}_2$ and $\mathcal{L}_3$. Incorporating $\mathcal{L}_2$ significantly boosted the discriminative performance of our method as the loss function is built upon the approximation of the concordance. In comparison, the network trained only with $\mathcal{L}_2$ provided discriminative performance which is not consistent over tested prediction and evaluation times since it does not consider the overall joint distributions of survival times and competing events. In addition, combining $\mathcal{L}_3$ led to a higher discriminative power by regularizing parameters of the shared subnetwork to ensure that the learned representations contain suitably rich information to perform step-ahead prediction.

## D    DETAILS OF THE DATASETS

For all the dataset considered in our experiments, missing values were replaced by zero-order hold interpolation and the ones that are still missing after the interpolation are imputed by the mean and mode for continuous and binary covariates, respectively.

### D.1    CF DATASET

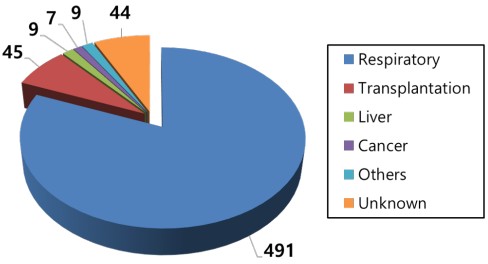

Figure 5: Death causes in the CF dataset.

Out of 10,995 patients, experiments were conducted on 5,883 adult patients with total of 90 features (11 static covariates and 79 time-varying covariates) whose follow-up data was available from January $1^{\text{st}}$ 2009 to December $31^{\text{st}}$ 2015. The covariates for individual CF patients include the followings: demographics, genetic mutations, lung function scores, hospitalization, bacterial lung infections, comorbidities, and therapeutic management. Of the 5,883 patient, 605 patients (10.28%) were followed until death and the remaining 5,278 patients (89.72%) were right-censored. Complications due to transplantation (7.43%) and CF-associated liver disease (1.49%) were the two most frequent causes of death after the respiratory failure. The detailed number of death causes in CF patients are illustrated in Figure 5. For each patient, longitudinal measurements were conducted roughly every year; the time interval between two adjacent measurements ranges from 0 to 69 months with mean of 9.20 months. Here, we discretized the time with a resolution of one month since the date information in the data was mostly available in month format. The number of yearly follow-ups was from 1 to 7 with mean of 5.34 measurements per patients.

### D.2    PBC DATASET

This data is from the Mayo Clinic trial in primary biliary cirrhosis (PBC) of the liver conducted between 1974 and 1984, which comprises of 312 patients with 15 follow-up variables (2 static covariates and 13 time-varying covariates)[12]. Among 312 PBC patients, 169 patients (54.17%) were followed until the competing events of our interest and the remaining 143 patients (45.83%) were right-censored. Two competing events are considered in the experiment: i) 140 (44.87%) deaths due to liver failures and ii) 29 (9.29%) patients who underwent liver transplantation. Throughout the experiments, all patients are aligned based on the start of the clinical study on PBC. The time interval between two adjacent measurements ranges from 1 to 69 months with mean of 10.69 months. Here,

---

[12]https://www.rdocumentation.org/packages/joineRML/versions/0.4.1/topics/pbc2

we discretized the time with a resolution of one month. The number of yearly follow-ups was from 1 to 16 with mean of 6.23 measurements per patients.

### D.3 ADNI DATASET

The Alzheimer's Disease Neuroimaging Initiative (ADNI) study data is a comprehensive dataset that tracks the progression of the Alzheimer's Disease (AD). In our experiment, we focused on 1,348 patients with 21 follow-up variables (4 static covariates and 17 time-varying covariates) and treated transition to Dementia as the even of our interest. Covariates include positron emission tomography (PET) regions of interest (ROI) scans, Magnetic Resonance (MRI) and diffusion tensor imaging (DTI), CSF and blood biomarkers, genetics, cognitive tests (ADAS-Cog), demographic and others. The time interval between two adjacent measurements ranges from 2 to 11 months with mean of 5.19 months. Here, we discretized the time with a resolution of one month. The number of yearly follow-ups was from 2 to 23 with mean of 8.65 measurements per patients.

## E    ADDITIONAL RESULTS WITH SINGLE RISK

We reported the discriminative performance of survival models on the ADNI dataset in Table 6, where we focus on a single risk. Our model achieves significant performance gain for the most of the the tested predication times (in year) and evaluation times (in year) in comparison with state-of-the-art methods. We confirm that Dynamic-DeepHit outperforms conventional methods not only in the setting with competing risks but also with a single risk.

Table 6: Comparison of $C_k(t, \Delta t)$ (mean $\pm$ std) for the ADNI dataset. Higher the better.

| Algorithms | | Dementia | | |
|---|---|---|---|---|
| | | $\Delta t = 1$ | $\Delta t = 3$ | $\Delta t = 5$ |
| $t = 4$ | cs-Cox | 0.828±0.30 | 0.827±0.30 | 0.827±0.30 |
| | RSF | 0.934±0.02 | 0.924±0.02 | 0.921±0.01 |
| | DeepHit | 0.928±0.01 | 0.925±0.00 | 0.921±0.01 |
| | JM | 0.885±0.02 | 0.884±0.02 | 0.883±0.02 |
| | Dynamic-Exp | 0.914±0.01 | 0.910±0.01 | 0.901±0.01 |
| | Dynamic-DeepHit | **0.952±0.01** | **0.940±0.03** | **0.948±0.02** |
| $t = 6$ | cs-Cox | **0.935±0.01** | 0.934±0.01 | 0.932±0.01 |
| | RSF | 0.905±0.02 | 0.903±0.02 | 0.902±0.01 |
| | DeepHit | 0.927±0.02 | 0.925±0.02 | 0.922±0.01 |
| | JM | 0.879±0.03 | 0.889±0.02 | 0.872±0.03 |
| | Dynamic-Exp | 0.898±0.01 | 0.888±0.02 | 0.875±0.02 |
| | Dynamic-DeepHit | 0.905±0.05 | **0.950±0.02** | **0.954±0.01** |

## F    ADDITIONAL RESULTS WITH THREE COMPETING RISKS

In this subsection, we reported the discriminative performance of survival models on the CF dataset with three competing risks in Table 7, where deaths from other causes (114 patients) are further categorized into deaths from transplant complications (45 patients) and those from other causes (69 patients) – this additional simulation is for the sake of illustrating how well Dynamic-DeepHit can scale to multiple (more than two) competing risks.

As seen in the table, our model achieves significant performance gain in discriminating the risks for all the the tested predication times (in year) and evaluation times (in year) in comparison with state-of-the-art methods. In particular, the gain over the landmarking methods were substantial for some specific predictions times, i.e., $t = 30$ for deaths from transplant complications and $t = 50$ for those from other causes. Although the landmarking methods are built upon relatively simple underlying survival models (i.e., cs-Cox and RSF), they become easily over-fitted since the number of events (for each cause) among patients at risks at these prediction times is small. (Please refer to the description of landmarking implementation in Appendix B.) Contrarily, the methods which incorporate patients with time-varying covariates at different time horizons, i.e., JM, Dynamic-Exp, and Dynamic-DeepHit, provided more consistent discriminative performance over different prediction times.

Table 7: Comparison of $C_k(t, \Delta t)$ (mean $\pm$ std) for the CF dataset with three competing risks. Higher the better.

| Algorithms | | Respiratory Failure | | |
|---|---|---|---|---|
| | | $\Delta t = 1$ | $\Delta t = 3$ | $\Delta t = 5$ |
| $t = 30$ | cs-Cox | 0.748±0.10 | 0.748±0.09 | 0.748±0.09 |
| | RSF | 0.926±0.01 | 0.921±0.01 | 0.919±0.01 |
| | JM | 0.833±0.02 | 0.878±0.01 | 0.870±0.01 |
| | Dynamic-Exp | 0.900±0.02 | 0.893±0.02 | 0.893±0.02 |
| | cs-Dynamic-DeepHit | 0.947±0.01 | **0.940±0.01** | 0.937±0.01 |
| | Dynamic-DeepHit | **0.948±0.01** | 0.939±0.01 | **0.938±0.01** |
| $t = 40$ | cs-Cox | 0.745±0.04 | 0.745±0.04 | 0.745±0.04 |
| | RSF | 0.887±0.02 | 0.885±0.03 | 0.881±0.03 |
| | JM | 0.858±0.02 | 0.872±0.01 | 0.884±0.01 |
| | Dynamic-Exp | 0.880±0.01 | 0.879±0.01 | 0.878±0.01 |
| | cs-Dynamic-DeepHit | 0.827±0.06 | 0.841±0.06 | 0.838±0.05 |
| | Dynamic-DeepHit | **0.947±0.02** | **0.950±0.01** | **0.949±0.01** |
| $t = 50$ | cs-Cox | 0.801±0.11 | 0.801±0.11 | 0.801±0.11 |
| | RSF | 0.893±0.01 | 0.887±0.02 | 0.884±0.01 |
| | JM | 0.878±0.02 | 0.884±0.01 | 0.889±0.01 |
| | Dynamic-Exp | 0.868±0.01 | 0.866±0.01 | 0.866±0.01 |
| | cs-Dynamic-DeepHit | 0.944±0.02 | 0.938±0.02 | 0.937±0.02 |
| | Dynamic-DeepHit | **0.954±0.01** | **0.960±0.00** | **0.959±0.00** |

| Algorithms | | Transplant Complication | | |
|---|---|---|---|---|
| | | $\Delta t = 1$ | $\Delta t = 3$ | $\Delta t = 5$ |
| $t = 30$ | cs-Cox | 0.554±0.21 | 0.553±0.21 | 0.555±0.21 |
| | RSF | 0.529±0.14 | 0.659±0.06 | 0.608±0.05 |
| | JM | 0.764±0.03 | 0.782±0.05 | 0.790±0.03 |
| | Dynamic-Exp | 0.749±0.09 | 0.748±0.09 | 0.746±0.09 |
| | cs-Dynamic-DeepHit | 0.929±0.01 | 0.926±0.01 | 0.926±0.01 |
| | Dynamic-DeepHit | **0.931±0.03** | **0.936±0.03** | **0.933±0.03** |
| $t = 40$ | cs-Cox | 0.608±0.09 | 0.608±0.09 | 0.609±0.09 |
| | RSF | 0.804±0.09 | 0.752±0.08 | 0.742±0.06 |
| | JM | 0.800±0.04 | 0.794±0.05 | 0.792±0.04 |
| | Dynamic-Exp | 0.759±0.07 | 0.759±0.07 | 0.759±0.07 |
| | cs-Dynamic-DeepHit | 0.887±0.05 | 0.881±0.05 | 0.883±0.05 |
| | Dynamic-DeepHit | **0.905±0.06** | **0.938±0.02** | **0.938±0.02** |
| $t = 50$ | cs-Cox | 0.815±0.09 | 0.815±0.09 | 0.815±0.09 |
| | RSF | 0.707±0.10 | 0.749±0.06 | 0.749±0.06 |
| | JM | 0.805±0.05 | 0.799±0.05 | 0.798±0.05 |
| | Dynamic-Exp | 0.747±0.07 | 0.747±0.07 | 0.747±0.07 |
| | cs-Dynamic-DeepHit | **0.908±0.01** | 0.920±0.01 | 0.921±0.01 |
| | Dynamic-DeepHit | 0.899±0.08 | **0.932±0.02** | **0.930±0.02** |

| Algorithms | | Other Cause | | |
|---|---|---|---|---|
| | | $\Delta t = 1$ | $\Delta t = 3$ | $\Delta t = 5$ |
| $t = 30$ | cs-Cox | 0.820±0.14 | 0.819±0.14 | 0.819±0.14 |
| | RSF | 0.809±0.27 | 0.805±0.24 | 0.795±0.24 |
| | JM | 0.703±0.03 | 0.700±0.04 | 0.752±0.05 |
| | Dynamic-Exp | 0.883±0.06 | 0.883±0.06 | 0.883±0.06 |
| | cs-Dynamic-DeepHit | 0.946±0.01 | 0.945±0.02 | 0.943±0.01 |
| | Dynamic-DeepHit | **0.953±0.03** | **0.945±0.02** | **0.944±0.02** |
| $t = 40$ | cs-Cox | 0.893±0.05 | 0.893±0.05 | 0.893±0.05 |
| | RSF | 0.781±0.04 | 0.791±0.04 | 0.791±0.04 |
| | JM | 0.744±0.03 | 0.718±0.05 | 0.711±0.02 |
| | Dynamic-Exp | 0.872±0.05 | 0.872±0.05 | 0.872±0.05 |
| | cs-Dynamic-DeepHit | 0.878±0.06 | 0.883±0.05 | 0.880±0.05 |
| | Dynamic-DeepHit | **0.927±0.02** | **0.934±0.02** | **0.934±0.02** |
| $t = 50$ | cs-Cox | 0.431±0.13 | 0.431±0.13 | 0.431±0.13 |
| | RSF | 0.689±0.06 | 0.637±0.12 | 0.637±0.12 |
| | JM | 0.733±0.02 | 0.708±0.04 | 0.706±0.04 |
| | Dynamic-Exp | 0.863±0.05 | 0.863±0.05 | 0.863±0.05 |
| | cs-Dynamic-DeepHit | 0.943±0.01 | 0.941±0.02 | **0.942±0.01** |
| | Dynamic-DeepHit | **0.945±0.02** | **0.943±0.01** | 0.939±0.02 |

When compared with JM and Dynamic-Exp which based on the underlying survival processes (the Cox proportional hazard model and the Exponential distribution, respectively), our proposed method provided significant improvement as it does not make assumptions about the underlying survival process and directly learns the joint distribution of the first hitting time and competing events. Furthermore, for most of the tested prediction and evaluation times, Dynamic-DeepHit outperformed the same network which was trained in a cause-specific fashion – the network is trained to learn the distribution of the first hitting time for each cause by treating the other causes as a form of right-censoring. These improvements imply that our network benefits from directly learning the joint distribution and scales well to multiple causes.

## G    ADDITIONAL RESULTS ON DYNAMIC RISK PREDICTION

We illustrate dynamic risk predictions and the trajectory of the hidden states for representative patients of the CF dataset in Figure 6 for a patient who are right-censored. The predicted risks for the patient in Figure 6 were low compared to those for the patient in Figure 3 or 4 in the manuscript. This is presumably because the patient had decreasing IV ABX hosp. and stable $FEV_1\%$ predicted. We confirmed that the trajectory of the hidden states in the shared network for the patient in Figure 6 moved toward "low risk" respiratory failure region.

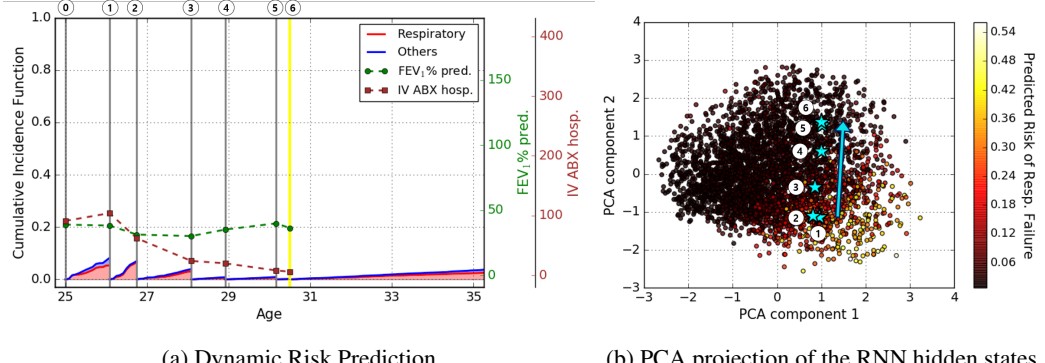

(a) Dynamic Risk Prediction                    (b) PCA projection of the RNN hidden states

Figure 6: Illustration of (a) dynamic risk predictions and (b) PCA projection of the RNN hidden states for a CF patient were right-censored. (a) The gray solid lines indicates the time at which new measurements are collected and the yellow solid line denotes the time at which the patient is right-censored. (b) The blue stars denote the corresponding PCA projection of the hidden states for the same patient.

## H    CAN DYNAMIC-DEEPHIT FLEXIBLY HANDLE MISSING DATA?

Table 8: Comparison of $C_k(t, \Delta t)$ (mean $\pm$ std) for the CF dataset with and without missing indicators (MI). Higher the better.

| Algorithms | | Resp. Failure | | | Other Causes | | |
|---|---|---|---|---|---|---|---|
| | | $\Delta t = 1$ | $\Delta t = 3$ | $\Delta t = 5$ | $\Delta t = 1$ | $\Delta t = 3$ | $\Delta t = 5$ |
| $t = 30$ | Proposed w/o MI | 0.946±0.01 | 0.940±0.01 | 0.939±0.01 | 0.926±0.02 | 0.919±0.03 | 0.913±0.03 |
| | Proposed w/ MI | **0.948±0.01** | **0.942±0.01** | **0.941±0.02** | **0.929±0.02** | **0.926±0.03** | **0.924±0.02** |
| $t = 40$ | Proposed w/o MI | 0.944±0.03 | 0.954±0.01 | 0.954±0.01 | 0.920±0.02 | 0.918±0.03 | 0.921±0.02 |
| | Proposed w/ MI | **0.957±0.02** | **0.958±0.02** | **0.958±0.02** | **0.935±0.02** | **0.934±0.03** | **0.934±0.03** |
| $t = 50$ | Proposed w/o MI | 0.958±0.01 | 0.959±0.01 | 0.959±0.01 | **0.934±0.02** | **0.939±0.02** | 0.938±0.02 |
| | Proposed w/ MI | **0.962±0.02** | **0.961±0.02** | **0.961±0.01** | 0.932±0.02 | 0.938±0.03 | **0.941±0.03** |

In longitudinal clinical studies, a survival model will have added value in practice if it can flexibly handle missing data to investigate the effect of missing measurements. (This is ubiquitous in medicine where not every patient undergo the exact same tests for measurements.) Fortunately, we can easily extend Dynamic-DeepHit to handle the missing data by providing missing indicators as auxiliary inputs along with time-varying covariates. More specifically, we can redefine the input sequence for patient $i$ as $\mathcal{X}^i = (\mathbf{X}^i, \mathbf{M}^i, \Delta^i)$ where $\mathbf{M}^i = \{\mathbf{m}_1^i, \cdots, \mathbf{m}_{M^i}^i\}$ is a sequence of mask vectors that indicate which covariates are missing at each time stamp. Here, $\mathbf{m}_j^i = [m_{j,1}^i, \cdots, m_{j,d_x}^i]$ for $j = 1, \cdots, M^i$ with $m_{j,d}^i = 1$ if the $d$-th time-varying covariate at time stamp $j$, $x_{j,d}^i$, is missing and $m_{j,d}^i = 0$, otherwise. Also, we need to slightly modify $\zeta(\mathbf{x}_{j+1}, \mathbf{y}_j)$ in the prediction loss (5) to only account for time-varying covariates that are available (we omit the dependency on $i$ for ease of notation):

$$\zeta(\mathbf{x}_{j+1}, \mathbf{y}_j) = \sum_{d \in \mathcal{I}} (1 - m_{j+1,d}) \cdot \zeta_d(x_{j+1,d}, \ y_{j,d}), \tag{8}$$

where $y_{j,d}$ is the step-ahead prediction at time stamp $j$ for the $d$-th time-varying covariate at time stamp $j + 1$, i.e., $x_{j+1,d}$.

In Table 8, we reported $C_k(t, \Delta t)$ for the CF dataset with two competing events for the proposed network with and without missing indicators. Two approaches showed very similar performance while the proposed network with missing indicators slightly outperformed the one without the missing indicators. We expect the gain mainly comes from the additional information about which covariates are missing and when they are missing. We leave further investigation on different scenarios in the missing data, e.g., missing at completely random and missing not at random, for future work.

