# OpenReview forum: "A Deep Learning Approach for Dynamic Survival Analysis with Competing Risks"
_ICLR.cc/2019/Conference_

### Official Review · AnonReviewer3 · 2018-11-01
**Good Empirical Performance, Questionable Scalability**

**Rating:** 4
**Confidence:** 3

**Review:**

The paper proposes a deep architecture that conducts survival analysis from longitudinal data where multiple competing risks are present. Experimental results demonstrate the effectiveness of the proposed method.  Specific comments follow:

1. A primary concern in the reviewer's opinion is the scalability of the architecture. While the reviewer appreciates the discussion of the scalability issue in terms of the output layer in the paper, the architecture might also not scalable if the number of competing risks is large because of the increase of the cause-specific subnetworks in the architecture.  Overall, the reviewer finds the lack of a principled approach to deal with competing risks and long time horizon presented in the paper.  Since dealing with competing risks in survival analysis is the goal of the paper, the reviewer finds the method presented insufficient for acceptance. As a remedy, for example, for the output layer, can the author consider the use of a neural net to model o_k at a particular time using time and f_{c_k}() as input?

Other issues:
2. it will be nice to explain (1) and (2) a little after presenting the formula.
3. page 5, $\mathbf{y}_j$ should also be explained because the next place where $\mathbf{y}_j$ is present is (4), which is one page later.
4. the term "dynamic survival analysis" is also obscure. What exactly does "dynamic" mean? To the reviewer's understanding, compared to standard survival analysis, the architecture models directly from raw longitudinal data of repeated measurements, and hence is called "dynamic".
5. even the "dynamic" part of the dynamic survival analysis is not very novel, see, for example,
Recurrent Marked Temporal Point Processes: Embedding Event History to Vector
and the follow-up works in the use of deep learning for point process modeling.

===============After Reading Authors' Response ================
The reviewer would like to thank the authors for their detailed response and careful revision of the paper to address the reviewer's concern. However, the reviewer is not persuaded by the authors' response. Specifically,

1. the reviewer is not satisfied with the explanation and modification to address the scalability issue stemming from both the cause-specific subnetworks and the output layer. Simplifying the structure and parameterization of cause-specific subnetworks when many are present seems like a comprise rather than a principled approach to address the issue. The same is true for the exponentially distributed parameterization of the output layer.

2. It is the reviewer's impression that for point process neural networks, it is possible to use the covariate information for prediction, as opposed to the claim given by the author.

---

> ### Author Response · Authors · 2018-11-16
> **Re: Good Empirical Performance, Questionable Scalability  (1)**
>
> We thank the reviewer for the valuable comments. Please find the answers below:
>
> 1.
>  We believe that the scalability issue is less likely to happen in the cause-specific subnetworks since their number of layers and nodes are chosen from hyper-parameter optimization -- the scalability issue due to the increased number of competing events should have been mitigated by selecting a smaller number of layers and nodes to avoid overfitting.
> We thank the reviewer for the suggestion of utilizing a different output layer to address the reviewer’s concern. However, we devised the current architecture for accomplishing the following two objectives. First, in the survival analysis under competing risks, our interest is to estimate the cause-specific CIF in (1) which shows the cumulative failure rates over time due to a particular cause [1]. This differs from the conventional classification or regression problem since the estimated CIF must satisfy two unique characteristics: i) the CIF is a function of both covariates and time-to-event that outputs a probability value in [0,1] and ii) it is a non-decreasing function of time-to-event. Second, we avoid using (semi-)parametric model that might suffer from mis-specification issues and fully exploit the representational capacity of neural networks by directly estimating the joint distribution of the first hitting time and the competing events to estimate the CIF as defined in (3).
> To highlight the aforementioned points and how well our model can scale to multiple causes, we provided three additional results in the revised paper:
> i) results on the CF dataset by further categorizing the competing risks into three causes,
> ii) comparison with cause-specific versions of Dynamic-DeepHit, and
> iii) a parametric version of Dynamic-DeepHit by replacing the current output layer to model the underlying survival process. For detailed descriptions of the first two results, please refer to Answer 1 to Reviewer 2.
> For the parametric version of Dynamic-DeepHit which was motivated by the reviewer’s comment, we modified the current output layer to model the underlying survival process with the Exponential distribution. Owing to model specification, this parametric version greatly reduces the number of output nodes (to the number of parameters for defining the Exponential distribution for each cause) while having potential limitations due to model mis-specification. Please refer to Appendix B.3 for the detailed description.
> As seen in Table 7 in Appendix F of the revised paper, our network outperforms the benchmarks even when we categorized the competing events of the CF dataset into three causes. When further compared Dynamic-DeepHit with its cause-specific versions and its parametric version, our proposed network achieved performance gain for most of the tested prediction and evaluation times. This implies that our network benefits from jointly learning the distribution of first hitting times of competing events (without any assumption about the underlying survival process) and scales well to multiple causes. Please refer to Appendix F in the revised paper for details.
>
> Reference:
> [1] J. P. Fine and R. J. Gray, “A proportional hazards model for the subdistribution of a competing risk,” Journal of the American Statistical Association, 1999.

---

> ### Author Response · Authors · 2018-11-16
> **Re: Good Empirical Performance, Questionable Scalability (2)**
>
> We thank the reviewer for the valuable comments. Please find the answers below:
>
> 2.
> The cause-specific cumulative incidence function in (1) implies the probability that a particular event k^{*} occurs on or before time \tau^{*} given the history of longitudinal measurements \mathcal{X}^{*}. Thus, from (1), we can assess the cause-specific risk of a patient as a function of time given the longitudinal measurement of this patient. Similarly, (2) implies the survival probability of a patient with longitudinal measurements \mathcal{X}^{*} until \tau^{*} (i.e., no event occurs on or before time \tau^{*}). We clarified the explanation of (1) and (2) in the revised paper, accordingly.
>
> 3.
> \mathbf{y}_{j} is the output of the shared subnetwork at time stamp j which indicates the step-ahead estimate of time-varying covariates, i.e., \mathbf{x}_{j+1}. These step-ahead predictions are used in the prediction loss (i.e. \mathcal{L}_{3}) to regularize the shared subnetwork. We clarified the explanation in page y the explanation in page 6.
>
> 4.
> In the paper, we used the term “dynamic” to differentiate the survival analysis on the basis of the longitudinal data from the static survival analysis which makes survival predictions based only on the current covariates. As shown in Figure 4, the CIF in (1) can be updated when new measurements are collected while incorporating the previous longitudinal measurements.
>
> 5.
> We thank the reviewer for suggesting the comparison with the related work in [2] (hereafter, RMTPP). Although both methods utilize an RNN structure to model the nonlinear dependency over the history of information, they address very different problems.
> RMTPP is built upon the marked temporal counting process whose goal is to predict the time to the next event and the indicator (marker) of that event given the history of previous events and markers. More specifically, utilizing the RNN structure, RMTPP issues these predictions at time stamps only at which the event and marker information is available. Thus, this method is suitable for modeling recurrent events, not the first hitting event that needs to be conditioned on the covariates, not on the previous of events.
> On the other hand, Dynamic-DeepHit is based on the first hitting time analysis (a.k.a. the time-to-event analysis) given the longitudinal measurements (not the previous event indicators). To account for the longitudinal measurements, our network utilizes an RNN structure as an encoder such that the cause-specific subnetworks and the output layer make outcomes based on the last hidden state of the RNN structure (overall, a sequence to time-to-event architecture). To our best knowledge, this paper is the first to investigate a deep learning approach for longitudinal time-to-event data on the basis of competing risks.
>
> Reference:
> [2] N. Du et al., “Recurrent marked temporal point processes: embedding event history to vector,” KDD, 2016.

---

> ### Author Response · Authors · 2018-11-28
> **Re: Answers to the additional reviewer's response.**
>
> We thank the reviewer for the feedback.
>
> 2 . We do acknowledge that the point process-based approaches can utilize the covariate information (i.e., history of measurements) for prediction.
> We also agree that the point process-based approaches can be applied to the "first-hitting time" analysis by limiting the maximum number of event count to one.  However, since point processes predict the next event time given the history of previous events, it is a more natural choice for studying the occurrence of recurrent events rather than first-hitting time event, where no other events nor the recurrent event can be observed once an event occurs.

---

### Official Review · AnonReviewer1 · 2018-11-02
**Novel modelling framework for well-motivated research problem**

**Rating:** 8
**Confidence:** 4

**Review:**

The authors present a novel deep learning representation for jointly modelling longitudinal measurements and dynamic time-to-event analysis where there are competing risks for a given event. The authors incorporate patient-level historical data using an RNN which allows updating of individual-level (i.e. personalized) risk predictions as additional data points are collected. This method (Dynamic-DeepHit) makes no assumptions about the underlying stochastic processes. The authors further evaluate the clinical utility of these methods in terms of interpretability of variable importance and dynamic risk predictions.
The work is clearly structured and clearly articulate a well-motivated research problem. It is also extremely well-placed within the historical context of previous work done in survival modelling. The authors have carried out an extensive review of the literature showing the evolution as well as the strengths and weaknesses of these methods.
My main concern with this manuscript is the handling of missing data. In the context of this study, the evaluation of missing data was inadequately investigated. This is an important problem within the context of what the authors are trying to achieve. Although it may be outside the scope of the current manuscript, different assumptions regarding missing data should be investigated. For example, if missing data was correlated with a particular outcome or a particular covariate, then replacing missing values with interpolation or with the mean and mode would lead to biased estimates.

---

> ### Author Response · Authors · 2018-11-16
> **Re: Novel modelling framework for well-motivated research problem**
>
> We thank the reviewer for suggesting further investigation regarding missing data. Indeed, we can easily extend Dynamic-DeepHit to flexibly handle the missing measurements by modifying two parts of the proposed network. First, we utilize mask vectors, m_{j} for j = 1, \cdots, M, that indicate which covariates are missing, as an auxiliary input of the network along with corresponding covariate vectors, x_{j} for j = 1, \cdots, M. (This approach has been used to handle missing measurements in time-series data [1,2] although they focused on in-hospital setting where measurements are frequently observed.) Then, we backpropagate the prediction loss (i.e. the \matchal{L}_{3}) that only corresponds to the step-ahead predictions, y_{j}, for the covariates, x_{j+1}, for j = 1, \cdots, M-1, that are not missing. In Appendix H of the revised paper, we provided descriptions of how our network can be extended to flexibly handle the missing data and the performance comparison with and without these missing indicators.
>
> References
> [1] J. Yoon et al., “Deep Sensing: Active Sensing using Multidirectional Recurrent Neural Networks,” ICLR 2018.
> [2] Z. Che et al., “Recurrent Neural Networks for Multivariate Time Series with Missing Values,” arXiv preprint arXiv:1606.01865, 2016.

---

### Official Review · AnonReviewer2 · 2018-11-03
**Seems like a solid survival analysis work, but not a good fit for ICLR**

**Rating:** 4
**Confidence:** 4

**Review:**

Summary:
The authors propose Dynamic-DeepHit, a survival analysis framework for modeling longitudinal data with multiple competing risks. As opposed to previous works, Dynamic-DeepHit can model survival events (e.g. death, cancer relapse) which can be driven by multiple, potentially competing, underlying risks. The proposed model uses an RNN shared across multiple risks for processing past-to-recent measurements, and multiple feedforward nets that accept the most recent measurements and the hidden layer of shared RNN. Joint predictions (across time and competing risks) are made using a softmax layer. The model is tested on two datasets where Dynamic-DeepHit outperforms other baselines.

Pros:
- Detailed explanation of survival analysis formulation.
- Experiments across multiple aspects: prediction performance, explaining the variable importance, visualizing the RNN hidden states

Issues:
- As the selling point of this model is its ability to capture competing risks, it is not very convincing that the experiments were conducted with only two competing risks. Can Dynamic-DeepHit truly capture multiple competing risks?
- The prediction performance was measured by "cause-specific time-dependent concordance index", which is described by Eq.5. But Eq.5 alone does not intuitively explain what it is trying to measure.
- Mayo Clinic data also has two competing risks, but Table 2 only shows the prediction performance for one risk, with the justification "liver transplant prediction is not in our interest". For the thoroughness of the experiments, why not put the complete result?
- All other issues aside: I can see that the authors put considerable effort into this work. But the effort is mainly focused on survival analysis, rather than learning representations. The novelty of this work regarding learning representations seems limited to me, as opposed to the contribution on improving survival analysis & medical prediction. This work would be much better received if submitted to a more relevant venue.

---

> ### Author Response · Authors · 2018-11-16
> **Re: Seems like a solid survival analysis work, but not a good fit for ICLR**
>
> We thank the reviewer for the valuable comments. Please find the answers below:
>
> 1.
> To highlight the scalability of Dynamic-DeepHit to multiple events, we provided a new set of experiments in the revised paper that show how well our network handles competing events by illustrating the performance improvement over the benchmarks. We reported the results in Table 7 in Appendix F of the revised paper; they include the followings:
>
> i) We further categorized the death causes of the CF dataset into three: 1) respiratory failure, 2) complications due to lung transplant, and 3) other causes. (Note that the first submission included only two causes which were the respiratory failure and other causes.) Then, we provided the discriminative performance for these three causes.
>
> ii) We added comparisons with the cause-specific version of Dynamic-DeepHit, where the proposed network is trained in a cause-specific manner (as the same as it is described for cs-Cox in Section 4.2) – the network learns the distribution of the first hitting time for each cause by treating the other causes as a form of right-censoring.
>
> iii) We compared a parametric version of Dynamic-DeepHit by replacing the current output layer to model the underlying survival process with the Exponential distribution. (This parametric version is motivated from Question1 of Reviewer 3.) Owing to model specification, this parametric version greatly reduces the number of output nodes (to the number of parameters for defining the Exponential distribution for each cause) while having potential limitations due to model mis-specification. Please refer to Appendix B.3 for the detailed description.
>
> As seen in the table, our network outperforms the benchmarks even when we further categorized the competing events of the CF dataset into three causes. When compared to the cause-specific versions and the parametric version, Dynamic-DeepHit achieved performance gain for most of the tested prediction and evaluation times. This implies that our network benefits from jointly learning the distribution of first hitting times of competing events (without any assumption about the underlying survival process) and scales well to multiple causes. Please refer to Appendix F in the revised paper for details.
>
> 2.
> The “cause-specific time-dependent concordance index (C-index)” in (6) is based on the idea of concordance [1] – a patient who experiences an event earlier should have a higher risk than a patient who survived longer. In the longitudinal setting, we need to account for the time at which the risk prediction is issued (to capture which longitudinal measurements are used as inputs) and the time at which we evaluate the discriminative performance (to capture possible changes in the estimated CIFs over time). Thus, the C-index defined in (6) provides the discriminative performance that reflects these time dependencies.
>
> 3.
> In PBC dataset, we have two event labels: i) death from liver failure and ii) liver transplant. We considered receiving a liver transplant as a competing event of death from liver failure since it hinders the liver failure from being observed during the study. (This is common in clinical studies; a similar example can be found in [2].) In Table 2, we only provided the performance for the death from liver failure since our interest is to assess the risk of having liver failure, not to assess the probability of receiving a liver transplant.
>
>
> References:
> [1] F. E. Harrell et al., “Evaluating the Yield of Medical Tests,” Journal of the American Medical Association, 1982.
> [2] M. Noordzij et al., “When Do We Need Competing Risks Methods for Survival Analysis in Nephrology?,” Nephrol Dial Transplant, 2013.

---

### Meta-Review · Area_Chair1 · 2018-12-17
**metareview for dynamic survival analysis paper**

**Confidence:** 4
**Recommendation:** Reject

**Metareview:**

While there was disagreement on this paper, reviewers remained unconvinced about the scalability and novelty of the presented work. While it was universally agreed that many positive points exist in this paper, it is not yet ready for publication.